# A Survey on Out-of-Distribution Detection in NLP

**Hao Lang**                                                                  *hao.lang@alibaba-inc.com*
*Alibaba Group*

**Yinhe Zheng**[*]                                                            *zhengyinhe1@163.com*
*Alibaba Group*

**Yixuan Li**                                                                 *sharonli@cs.wisc.edu*
*Department of Computer Sciences, University of Wisconsin-Madison*

**Jian Sun**                                                                  *jian.sun@alibaba-inc.com*
*Alibaba Group*

**Fei Huang**                                                                 *f.huang@alibaba-inc.com*
*Alibaba Group*

**Yongbin Li**[*]                                                             *shuide.lyb@alibaba-inc.com*
*Alibaba Group*

**Reviewed on OpenReview:** *https://openreview.net/forum?id=nYjSkOy8ij*

## Abstract

Out-of-distribution (OOD) detection is essential for the reliable and safe deployment of machine learning systems in the real world. Great progress has been made over the past years. This paper presents the first review of recent advances in OOD detection with a particular focus on natural language processing approaches. First, we provide a formal definition of OOD detection and discuss several related fields. We then categorize recent algorithms into three classes according to the data they used: (1) OOD data available, (2) OOD data unavailable + in-distribution (ID) label available, and (3) OOD data unavailable + ID label unavailable. Third, we introduce datasets, applications, and metrics. Finally, we summarize existing work and present potential future research topics.

## 1 Introduction

Natural language processing systems deployed in the wild often encounter out-of-distribution (OOD) samples that are not seen in the training phase. For example, a natural language understanding (NLU) component in a functional dialogue system is typically developed using a limited training set that encompasses a finite number of intents. However, when deployed, the NLU component may be exposed to an endless variety of user inputs, some of which may include out-of-distribution intents not supported by the training. A reliable and trustworthy NLP model should not only obtain high performance on samples from seen distributions, i.e., In-distribution (ID) samples, but also accurately detect OOD samples (Amodei et al., 2016). For instance, when building task-oriented dialogue systems, it is hard, if not impossible, to cover all possible user intents in the training stage. It is critical for a practical system to detect these OOD intents or classes in the testing phase so that they can be properly handled (Zhan et al., 2021).

However, existing flourishes of neural-based NLP models are built upon the *closed-world assumption*, i.e., the training and testing data are sampled from the same distribution (Vapnik, 1991). This assumption is often violated in practice, where deployed models are generally confronting an *open-world*, i.e., some testing data

---

[*] Corresponding author.

may come from OOD distributions that are not seen in training (Bendale & Boult, 2015; Fei & Liu, 2016). It is also worth noting that although large language models (LLMs) have exhibited superior performance in various tasks by training on an enormous set of texts, the knowledge exhibited in these training texts is limited to a certain cut-off date. OOD detection is still an important task for these LLMs since the world is constantly involving. New tasks may be developed after the knowledge cut-off date.

A rich line of work has been proposed to tackle problems introduced by OOD samples. Specifically, distributional shifts in NLP can be broadly divided into two types: 1. semantic shift, i.e., OOD samples may come from unknown categories, and therefore should not be blindly predicted into a known category; 2. non-semantic shift, i.e., OOD samples may come from different domains or styles but share the same semantic with some ID samples (Arora et al., 2021). The detection of OOD samples with semantic shift is the primary focus of this survey, where the label set $\mathcal{Y}$ of ID samples is different from that of OOD samples. The ability to detect OOD samples is critical for building safe NLP systems for, say, text classification (Hendrycks & Gimpel, 2017), question answering (Kamath et al., 2020), and machine translation (Kumar & Sarawagi, 2019).

Although there already exists surveys on many aspects of OOD, such as OOD generalization (Wang et al., 2022) and OOD detection in computer vision (CV) (Yang et al., 2021), *a comprehensive survey for OOD detection in NLP is still lacking and thus urgently needed for the field.* Concretely, applying OOD detection to NLP tasks requires specific considerations, e.g., tackling discrete input spaces, handling complex output structures, and considering contextual information, which have not been thoroughly discussed. Our key contributions are summarized as follows:

**1.** We propose a novel taxonomy of OOD detection methods based on the availability of OOD data (Section 3) and discuss their pros and cons for different settings (Section 6.1).

**2.** We present a survey on OOD detection in NLP and identify various differences between OOD detection in NLP and CV (Section 6.3).

**3.** We review datasets, applications (Section 4), metrics (Section 5), and future research directions (Section 6.4) of OOD detection in NLP.

## 2 OOD Detection and Related Areas

**Definition 1** (Data distribution). *Let $\mathcal{X}$ denote a nonempty input space and $\mathcal{Y}$ a label (semantic) space. A data distribution is defined as a joint distribution $P(X, Y)$ over $\mathcal{X} \times \mathcal{Y}$. $P(X)$ and $P(Y)$ refer to the marginal distributions for inputs and labels, respectively.*

In practice, common non-semantic distribution shifts on $P(X)$ include domain shifts (Wang et al., 2022), sub-population shifts (Koh et al., 2021), or style changes (Pavlick & Tetreault, 2016; Duan et al., 2022). Typically, the label space $\mathcal{Y}$ remains unchanged in these non-semantic shifts, and sophisticated methods are developed to improve the model's robustness and generalization performance. On the contrary, semantic distribution shifts on $P(Y)$ generally lead to a new label space $\widetilde{\mathcal{Y}}$ that is different from the one seen in the training phase (Bendale & Boult, 2016). These shifts are usually caused by the occurrence of new classes at the testing stage. In this work, we mainly focus on detecting OOD samples with semantic shifts, the formal definition of which is given as follows:

**Definition 2** (OOD detection). *We are given an ID training set $\mathcal{D}_{train} = \{(\mathbf{x}_i, y_i)\}_{i=1}^{L} \sim P_{train}(X, Y)$, where $\mathbf{x}_i \in \mathcal{X}_{train}$ is a training instance, and $y_i \in \mathcal{Y}_{train} = \{1, 2, ..., K\}$ is the associated class label. Facing the emergence of unknown classes, we are given a test set $\mathcal{D}_{test} = \{(\mathbf{x}_i, y_i)\}_{i=1}^{N} \sim P_{test}(X, Y)$, where $\mathbf{x}_i \in \mathcal{X}_{test}$, and $y_i \in \mathcal{Y}_{test} = \{1, ..., K, K+1\}$. Note that class $K+1$ is a group of novel categories representative of OOD samples, which may contain more than one class. The overall goal of OOD detection is to learn a predictive function $f$ from $\mathcal{D}_{train}$ to achieve a minimum expected risk on $\mathcal{D}_{test}$: $\min_f \mathbb{E}_{(\mathbf{x}, y) \sim \mathcal{D}_{test}} \mathbb{I}(y \neq f(\mathbf{x}))$, i.e., not only classify known classes but also detect the unknown categories.*

We briefly describe the related research areas:

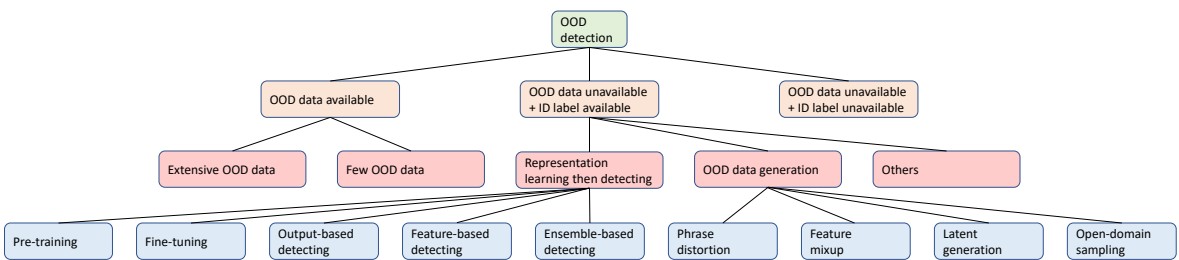

Figure 1: Taxonomy of OOD detection methods.

**Domain generalization** (DG) (Wang et al., 2022; Zhou et al., 2022a), or out-of-distribution generalization, aims to learn a model from one or several source domains and expect these learned models to generalize well on unseen testing domains (i.e., target domains). DG mainly focuses on the non-semantic drift, i.e., the training and testing tasks share the same label space $\mathcal{Y}$ while they have different distributions over the input space $\mathcal{X}$. Different from DG, OOD detection handles a different label space during testing.

**Domain adaptation** (DA) (Blitzer et al., 2006) follows most settings of DG except that DA has access to some unlabeled data from the target domain in the training process (Ramponi & Plank, 2020). Similar to DG, DA also assumes the label space remains unchanged.

**Zero-shot learning** (Wang et al., 2019) aims to use learned models to classify samples from unseen classes. The main focus of zero-shot learning is to obtain the correct labels for these unseen classes. However, OOD detection in general only needs to detect samples from unseen classes without further classifying them. Some OOD detection models can also classify samples from seen classes since these samples are annotated in the training set.

**Meta-learning** (Vilalta & Drissi, 2002) aims to learn from the model training process so that models can quickly adapt to new data. Different from meta-learning, achieving strong few-shot performance is not the major focus of OOD detection. Nevertheless, the idea of meta-learning can serve as a strategy for OOD detection (Xu et al., 2019; Li et al., 2021) by simulating the behaviors of predicting unseen classes in the training stage.

**Positive-unlabeled Learning** (Zhang & Zuo, 2008), or PU learning, aims to train a classifier with only positive and unlabeled examples while being able to distinguish both positive and negative samples in testing. However, OOD detection considers multiple classes in training. PU learning approaches can be applied to tackle the OOD detection problem when only one labeled class exists (Li & Liu, 2003).

**Transfer Learning** (Ruder et al., 2019) aims to leverage data from additional domains or tasks to train a model with better generalization properties. Most transfer learning approaches target at producing robust representations that are agnostic of their downstream tasks. OOD detection can be regarded as a downstream task for transfer learning.

## 3 Methodology

A major challenge of OOD detection is the lack of representative OOD data, which is important for estimating OOD distributions (Zhou et al., 2021b). As shown in Figure 1, we classify existing OOD detection methods into three categories according to the availability of OOD data. Methods covered in our survey are selected following the criteria listed in Appendix A.

### 3.1 OOD Data Available

Methods in this category assume access to both labeled ID and OOD data during training. Based on the quantity and diversity of OOD data, we further classify these methods into two subcategories:

### 3.1.1  Detection with Extensive OOD Data

Some methods assume that we can access extensive OOD data in the training process together with ID data. In this subcategory, one line of work formulates OOD detection as a discriminative classification task, i.e., a special label is allocated in the label space for OOD samples. Fei & Liu (2016); Larson et al. (2019) formed a $(K + 1)$-way classification problem, where $K$ denoted the number of ID classes and the $(K + 1)^{th}$ class represented OOD samples. Larson et al. (2019); Kamath et al. (2020) regarded OOD detection as a binary classification problem, where the two classes correspond to ID and OOD samples, respectively. Kim & Kim (2018) introduced a neural joint learning model with a multi-class classifier for domain classification and a binary classifier for OOD detection.

Another line of work optimizes an outlier exposure regularization term on these OOD samples to refine the representations and OOD scores learned by the OOD detector. Hendrycks et al. (2019a) introduced a generalized outlier exposure (OE) loss to train models on both ID and OOD data. For example, when using the maximum softmax probability detector (Hendrycks & Gimpel, 2017), the OE loss pushes the predicted distribution of OOD samples to a uniform distribution (Lee et al., 2018a). When the labels of ID data are not available, the OE loss degenerates to a margin ranking loss on the predicted distributions of ID and OOD samples. Zeng et al. (2021b) added an entropy regularization objective to enforce the predicted distributions of OOD samples to have high entropy.

### 3.1.2  Detection with Few OOD Data

Some methods assume that we can only access a small amount of OOD data besides ID data. This setting is more realistic in practice since it is expensive to annotate large-scale OOD data. Several methods in this subcategory are developed to generate pseudo samples based on a small number of seed OOD data. Chen & Yu (2021) constructed pseudo-labeled OOD candidates using samples from an auxiliary dataset and kept only the most beneficial candidates for training through a novel election-based filtering mechanism. Rather than directly creating OOD samples in natural language, Zeng et al. (2021b) borrowed the idea of adversarial attack (Goodfellow et al., 2014) to obtain model-agnostic worst-case perturbations in the latent space, where these perturbations or noise can be regarded as augmentations for OOD samples. Note that techniques used by these methods with few OOD data (i.e., increasing the diversity and quantity of OOD data) may also help the detection methods with extensive OOD data (Shu et al., 2021). See Section 4 and Appendix B for more details of OOD detection datasets.

## 3.2  OOD Data Unavailable + ID Label Available

Building OOD detectors using only labeled ID data is the major focus of research communities. We generally classify existing literature into three subcategories based on their learning principles:

### 3.2.1  Learn Representations Then Detect

Some methods formulize the OOD detector $f$ into two components: a representation extractor $g$ and an OOD scoring function $d$, i.e., $f(\mathbf{x}) = d(g(\mathbf{x}))$: $g$ aims to capture a representation space $\mathcal{H}$ in which ID and OOD samples are distinct, and $d$ maps each extracted representation into an OOD score so that OOD samples can be detected based on a selected threshold. We provide an overview of methods to enhance these two components:

**a.  Representation Learning** usually involves two stages: (1) a *pre-training* stage leverages massive unlabeled text corpora to extract representations that are suitable for general NLP tasks; (2) a *fine-tuning* stage uses labeled in-domain data to refine representations for specified downstream tasks. An overview of these two stages is given here:

**Pre-training** Pre-trained transformer models such as BERT (Kenton & Toutanova, 2019) have become the de facto standard to implement text representation extractors. Hendrycks et al. (2020) systematically measured the OOD detection performance on various representation extractors, including bag-of-words models, ConvNets (Gu et al., 2018), LSTMs (Hochreiter & Schmidhuber, 1997), and pre-trained transformer

models (Vaswani et al., 2017). Their results show that pre-trained models achieve the best OOD detection performance, while the performances of all other models are often worse than chance. The success of pre-trained models attributes to these diverse corpora and effective self-supervised training losses used in training (Hendrycks et al., 2019b).

Moreover, it is observed that better-calibrated models generally produce higher OOD detection performance (Lee et al., 2018a). Desai & Durrett (2020) evaluated the calibration of two pre-trained models, BERT and RoBERTa (Liu et al., 2019), on different tasks. They found that pre-trained models were better calibrated in out-of-domain settings, where non-pre-trained models like ESIM (Chen et al., 2017) were overconfident. Dan & Roth (2021) also demonstrated that larger pre-trained models are more likely to be better calibrated and thus result in higher OOD detection performance.

**Fine-tuning** With the help of labeled ID data, various approaches are developed to fine-tune the representation extractor to widen margins between ID and OOD samples. Lin & Xu (2019) proposed a large margin cosine loss (LMCL) to maximize the decision margin in the latent space. LMCL simultaneously maximizes inter-class variances and minimizes intra-class variances. Yan et al. (2020) introduced a semantic-enhanced Gaussian mixture model to enforce ball-like dense clusters in the feature space, which injects semantic information of class labels into the Gaussian mixture distribution.

Zeng et al. (2021a); Zhou et al. (2021b) leveraged contrastive learning to increase the discrepancy for representations extracted from different classes. They hypothesized that increasing inter-class discrepancies helps the model learn discriminative features for ID and OOD samples and therefore improves OOD detection performances. Concretely, a supervised contrastive loss (Khosla et al., 2020; Gunel et al., 2021) and a margin-based contrastive loss was investigated. Zeng et al. (2021b) proposed a self-supervised contrastive learning framework to extract discriminative representations of OOD and ID samples from unlabeled data. In this framework, positive pairs are constructed using the back-translation scheme. Zhou et al. (2022b) applied KNN-based contrastive learning losses to OOD detectors and Wu et al. (2022) used a reassigned contrastive learning scheme to alleviate the over-confidence issue in OOD detection. Cho et al. (2022) proposed a contrastive learning based framework that encourages intermediate features to learn layer-specialized representations. Mou et al. (2022) proposed to align representation learning with scoring function via unified neighborhood learning.

Moreover, there are some regularized fine-tuning schemes to tackle the over-confidence issue of neural-based OOD detectors. Kong et al. (2020) addressed this issue by introducing an off-manifold regularization term to encourage producing uniform distributions for pseudo off-manifold samples. Shen et al. (2021) designed a novel domain-regularized module that is probabilistically motivated and empirically led to a better generalization in both ID classification and OOD detection.

Recently, Uppaal et al. (2023) systematically explored the necessity of fine-tuning on ID data for OOD detection. They showed experimentally that pre-trained models without fine-tuning on the ID data outperform their fine-tuned counterparts with a variety of OOD detectors, when there is a significant difference between the distributions of ID and OOD data (e.g., ID and OOD data are sampled from different datasets).

**b. OOD Scoring** processes usually involve a scoring function $d$ to map the representations of input samples to OOD detection scores. A higher OOD score indicates that the input sample is more likely to be OOD. The scoring function can be categorized as (but not limited to) the following: (1) *output-based detecting*, (2) *feature-based detecting*, and (3) *ensemble-based detecting*:

**Output-based Detecting** compute the OOD score based on the predicted probabilities. Hendrycks & Gimpel (2017); Hendrycks et al. (2020) used the maximum Softmax probability as the detection score, and Liang et al. (2018) improved this scheme with the temperature scaling approach. Shu et al. (2017) employed $K$ 1-vs-rest Sigmoid classifiers for $K$ predefined ID classes and used the maximum probabilities from these classifiers as the detection score. Liu et al. (2020) proposed an energy score for better distinguishing ID/OOD samples. The energy score is theoretically aligned with the probability density of the inputs.

**Feature-based Detecting** leverages features derived from intermediate layers of the model to implement density-based and distance-based scoring functions. Gu et al. (2019) proposed a nearest-neighbor based method with a distance-to-measure metric. Breunig et al. (2000) used a local outlier factor as the detection

score, in which the concept "local" measured how isolated an object was with respect to surrounding neighborhoods. Lee et al. (2018b); Podolskiy et al. (2021) obtained the class-conditioned Gaussian distributions with respect to features of the deep models under Gaussian discriminant analysis. This scheme resulted in a confidence score based on the Mahalanobis distance. While Mahalanobis imposes a strong distributional assumption on the feature space, Sun et al. (2022) demonstrated the efficacy of non-parametric nearest neighbor distance for OOD detection. Zhang et al. (2021) proposed a post-processing method to learn an adaptive decision boundary (ADB) for each ID class. Specifically, the ADB is learned by balancing both the empirical and open space risks (Scheirer et al., 2014). Chen et al. (2022) proposed to average all token representations from each intermediate layer of pre-trained language models as the sentence embedding for better OOD detection. Recently, Ren et al. (2023) proposed to detect OOD samples for conditional language generation tasks (such as abstractive summarization and translation) by calculating the distance between testing input/output and a corresponding background model in the feature space.

**Ensemble-based Detecting** uses predictive uncertainty of a collection of supporting models to compute OOD scores. Specifically, an input sample is regarded as an OOD sample if the variance of these models' predictions is high. Gal & Ghahramani (2016) modeled uncertainties by applying dropouts to neural-based models. This scheme approximates Bayesian inference in deep Gaussian processes. Lakshminarayanan et al. (2017) used deep ensembles for uncertainty quantification, where multiple models with the same architecture were trained in parallel with different initializations. Lukovnikov et al. (2021) further proposed a heterogeneous ensemble of models with different architectures to detect compositional OOD samples for semantic parsing.

### 3.2.2 Generate Pseudo OOD Samples

A scheme to tackle the problem of lacking OOD training samples is to generate pseudo-OOD samples during training (Lang et al., 2022). With these generated pseudo-OOD samples, OOD detectors can be solved by methods designed for using both labeled ID and OOD data. There are mainly four types of approaches for generating pseudo-OOD samples: (1) *phrase distortion*, (2) *feature mixup*, (3) *latent generation*, and (4) *open-domain sampling*:

**Phrase Distortion** approaches generate pseudo-OOD samples for NLP tasks by selectively replacing text phrases in ID samples. Ouyang et al. (2021) proposed a data manipulation framework to generate pseudo-OOD utterances with importance weights. Choi et al. (2021) proposed OutFlip, which revised a white-box adversarial attack method HotFlip to generate OOD samples. Shu et al. (2021) created OOD instances from ID examples with the help of a pre-trained language model. Kim et al. (2023) constructed a surrogate OOD dataset by sequentially masking tokens related to ID classes.

**Feature Mixup** strategy (Zhang et al., 2018) is also a popular technique for pseudo data generation. Zhan et al. (2021) generated OOD samples by performing linear interpolations between ID samples from different classes in the representation space. Zhou et al. (2021a) leveraged the manifold Mixup scheme (Verma et al., 2019) for pseudo OOD sample generation. Intermediate layer representations of two samples from different classes are mixed using scalar weights sampled from the Beta distribution. These feature-mixup-based methods achieved promising performance while remaining conceptually and computationally straightforward.

**Latent Generation** approaches considered to use generative adversarial networks (GAN) (Goodfellow et al., 2020) to produce high-quality pseudo OOD samples. Lee et al. (2018a) proposed to generate boundary samples in the low-density area of the ID distribution as pseudo-OOD samples. Du et al. (2022b) proposed virtual outlier synthesis (VOS), showing that sampling low-likelihood outliers in the feature space is more tractable and effective than synthesizing images in the high-dimensional pixel space. Tao et al. (2023) extended VOS to the non-parametric setting, which enabled sampling latent-space outliers without making strong assumptions on the feature distributions. Ryu et al. (2018) built a GAN on ID data and used the discriminator to generate OOD samples in the continuous feature space. Zheng et al. (2020) generated pseudo-OOD samples using an auto-encoder with adversarial training in the discrete text space. Marek et al. (2021) proposed OodGAN, in which a sequential generative adversarial network (SeqGAN) (Yu et al., 2017) was used for OOD sample generation. This model follows the idea of Zheng et al. (2020) but works directly on

texts and hence eliminates the need to include an auto-encoder. Very recently, Du et al. (2023) demonstrated synthesizing photo-realistic high-quality outliers by leveraging advanced diffusion-based models (Rombach et al., 2022).

**Open-domain Sampling** approaches directly use sentences from other corpora as pseudo-OOD samples (Zhan et al., 2021).

### 3.2.3 Other Methods

We also review some representative methods that do not belong to the above two categories. Vyas et al. (2018) proposed to use an ensemble of classifiers to detect OOD, where each classifier was trained in a self-supervised manner by leaving out a random subset of training data as OOD data. Li et al. (2021) proposed $k$Folden, which included $k$ classifiers for $k$ class labels. Each classifier was trained on a subset with $k - 1$ classes while leaving one class unknown. Zhou et al. (2023) also proposed an OOD training method based on ensemble methods. Wu et al. (2023) proposed a novel multi-level knowledge distillation-based approach for OOD detection. Tan et al. (2019) tackled the problem of OOD detection with limited labeled ID training data and proposed an OOD-resistant Prototypical Network to build the OOD detector. Ren et al. (2019); Gangal et al. (2020) used the likelihood ratio produced by generative models to detect OOD samples. The likelihood ratio effectively corrects confounding background statistics for OOD detection. Ryu et al. (2017) employed the reconstruction error as the detection score. Ouyang et al. (2023) proposed an unsupervised prefix-tuning-based OOD detection framework to be lightweight.

## 3.3 OOD data unavailable + ID label unavailable

OOD detection using only unlabeled ID data can be used for non-classification tasks. In fact, when ID labels are unavailable, our problem setting falls back to the classical anomaly detection problem, which is studied under a rich set of literature (Chalapathy & Chawla, 2019; Pang et al., 2021). However, this problem setting is rarely investigated in NLP studies. We keep this category here for the completeness of our survey while leaning most of our focus on NLP-related works.

Methods in this category mainly focus on extracting more robust features and making a more accurate estimation for the data distribution. For example, Zong et al. (2018) proposed a DAGMM model for unsupervised OOD detection, which utilized a deep auto-encoder to generate low-dimensional representations to estimate OOD scores. Xu et al. (2021) transformed the feature extracted from each layer of a pre-trained transformer model into one low-dimension representation based on the Mahalanobis distance, and then optimized an OC-SVM for detection. Some works also use language models (Nourbakhsh & Bang, 2019) and word representations (Bertero et al., 2017) to detect OOD inputs on various tasks such as log analysis (Yadav et al., 2020) and data mining (Agrawal & Agrawal, 2015).

## 4 Datasets and Applications

In this section, we briefly discuss representative datasets and applications for OOD detection. We classify existing OOD detection datasets into three categories according to the construction schemes of OOD samples in the testing stage:

**(1) Annotate OOD Samples:** This category of datasets contains OOD samples that are manually annotated by crowd-source workers. Specifically, CLINIC150 (Larson et al., 2019) is a manually labeled single-turn dialogue dataset that consists of 150 ID intent classes and 1,200 out-of-scope queries. STAR (Mosig et al., 2020) is a multi-turn dialogue dataset with annotated turn-level intents, in which OOD samples are labeled as "out_of_scope", "custom", or "ambiguous". ROSTD (Gangal et al., 2020) is constructed by annotating about 4,000 OOD samples on the basis of the dataset constructed by Schuster et al. (2019).

**(2) Curate OOD samples using existing classes:** This category of datasets curates OOD examples by holding out a subset of classes in a given corpus (Zhang et al., 2021). Any text classification datasets can be adopted in this process.

**(3) Curate OOD samples using other corpora:** This category of datasets curates OOD samples using samples extracted from other datasets (Hendrycks et al., 2020; Zhou et al., 2021b), i.e., samples from other corpora are regarded as OOD samples. In this way, different NLP corpora can be combined to construct OOD detection tasks.

OOD detection tasks have also been widely applied in various NLP applications. We generally divide these applications into two types:

**(1) Classification Tasks** are natural applications for OOD detectors. Almost every text classifier built in the closed-world assumption needs the OOD detection ability before deploying to production. Specifically, intent classification for dialogue systems is the most common application for OOD detection (Larson et al., 2019; Lin & Xu, 2019). Other popular application scenarios involve general text classification (Zhou et al., 2021b; Li et al., 2021), sentiment analysis (Shu et al., 2017), and topic prediction (Rawat et al., 2021).

**(2) Selective Prediction Tasks** predict higher-quality outputs while abstaining on uncertain ones (Geifman & El-Yaniv, 2017; Varshney et al., 2022). This setting can be combined naturally with OOD detection techniques. A few studies use OOD detection approaches for selective prediction in question answering, semantic equivalence judgments, and entailment classification (Kamath et al., 2020; Xin et al., 2021).

## 5 Metrics

The main purposes of OOD detectors are separating OOD and ID input samples, which is essentially a binary classification process. Most methods mentioned above try to compute an *OOD score* for this problem. Therefore, threshold-free metrics that are generally used to evaluate binary classifiers are commonly used to evaluate OOD detectors:

**AUROC**: Area Under the Receiver Operating Characteristic curve (Davis & Goadrich, 2006). The Receiver Operating Characteristic curve is a plot showing the true positive rate $TPR = \frac{TP}{TP+FN}$ and the false positive rate $FPR = \frac{FP}{FP+TN}$ against each other, in which TP, TN, FP, FN denotes true positive, true negative, false positive, false negative, respectively. For OOD detection tasks, ID samples are usually regarded as positive. Specifically, a random OOD detector yields an AUROC score of 50% while a "perfect" OOD detector pushes this score up to 100%.

**AUPR**: Area Under the Precision-Recall curve (Manning & Schutze, 1999). The Precision-Recall curve plots the precision $\frac{TP}{TP+FP}$ and recall $\frac{TP}{TP+FN}$ against each other. The metric AUPR is used when the positive and negative classes in the testing phase are severely imbalanced because the metric AUROC is biased in this situation. Generally, two kinds of AUPR scores are reported: 1) **AUPR-IN** where ID samples are specified as positive; 2) **AUPR-OUT** where OOD samples are specified as positive.

Besides these threshold-free metrics, we are also interested in the performance of OOD detectors after the deployment, i.e., when a specific threshold is selected. The following metric is usually used to measure this performance:

**FPR@$N$**: The value of FPR when TPR is $N\%$ (Liang et al., 2018; Lee et al., 2018a). This metric measures the probability that an OOD sample is misclassified as ID when the TPR is at least $N\%$. Generally, we set $N = 95$ or $N = 90$ to ensure high performance on ID samples. This metric is important for a deployed OOD detector since obtaining a low FPR score while achieving high ID performance is important for practical systems.

In addition to the ability to detect OOD samples, some OOD detectors are also combined with downstream ID classifiers. Specifically, for a dataset that contains $K$ ID classes, these modules allocate an additional OOD class for all the OOD samples and essentially perform a $K + 1$ class classification task. The following metrics are used to evaluate the overall performance of these modules:

**F1**: The macro F1 score is used to evaluate classification performance, which keeps the balance between precision and recall. Usually, F1 scores are calculated over all samples to estimate the overall performance. Some studies also compute F1 scores over ID and OOD samples, respectively, to evaluate fine-grained performances (Zhang et al., 2021).

**Acc**: The accuracy score is also used to evaluate classification performance (Zhan et al., 2021). See Appendix C for more details of various metrics.

## 6  Discussion

### 6.1  Pros and Cons for Different Settings

Labeled OOD data provide valuable information for OOD distributions, and thus models trained using these OOD samples usually achieve high performance in different applications. However, the collection of labeled OOD samples requires additional efforts that are extremely time-consuming and labor-intensive. Moreover, due to the infinite compositions of language, it is generally impractical to collect OOD samples for all unseen cases. Using only a small subset of OOD samples may lead to serious selection bias issues and thus hurt the generalization performance of the learned OOD detector. Therefore, it is important to develop OOD detection methods that do not rely on labeled OOD samples.

OOD detection using only labeled ID data fits the above requirements. The representation learning and detecting approaches decompose the OOD detection process in this setting into two stages so that we can separately optimize each stage. Specifically, the representation learning stage attempts to learn distinct feature spaces for ID/OOD samples. Results show that this stage benefits from recent advances in pretraining and semi-supervised learning schemes on unlabeled data. OOD scoring functions aim to produce reliable scores for OOD detection. Various approaches generate the OOD score with different distance measurements and distributions. Another way to tackle the problem of lacking annotated OOD data is to generate pseudo-OOD samples. Approaches in this category benefit from the strong language modeling prior and the generation ability of pre-trained models.

In some applications, we can only obtain a set of ID data without any labels. This situation is commonly encountered in non-classification tasks where we also need to detect OOD inputs. Compared to NLP, this setting is more widely investigated in other fields like machine learning and computer vision (CV). Popular approaches involve using estimated distribution densities or reconstruction losses as the OOD scores.

### 6.2  Large Language Models for OOD Detection

Recent progress in large language models (LLMs) has led to quality approaching human performance on research datasets and thus LLMs dominate the NLP field (Brown et al., 2020; Bommasani et al., 2021). With LLMs, many NLP tasks such as text summarization, semantic parsing, and translation can be formulated as a general "text to text" task and have achieved promising results (Raffel et al., 2020; Zhang et al., 2020). In this setting, OOD samples are assumed to be user inputs that significantly deviate from the training data distribution (Xu et al., 2021; Lukovnikov et al., 2021; Ren et al., 2023; Vazhentsev et al., 2023). These OOD inputs should also be detected because many machine learning models can make overconfident predictions for OOD inputs, leading to significant AI safety issues (Hendrycks & Gimpel, 2017; Ovadia et al., 2019). Moreover, language models are typically trained to classify the next token in an output sequence and may suffer even worse degradation on OOD inputs as the classification is done auto-regressively over many steps. Hence, it is important to know when to trust the generated output of LLMs (Si et al., 2023).

In parallel, LLMs embed broad-coverage world knowledge that can help a variety of downstream tasks (Petroni et al., 2019). Recently, Dai et al. (2023b) apply world knowledge from LLMs to multimodal OOD detection (Ming et al., 2022) by generating descriptive features for ID class names (Menon & Vondrick, 2023), which significantly increase the OOD detection performance. Meanwhile, LLMs can be explored for text data augmentation generally (Dai et al., 2023a), which could enhance OOD performance by generating diverse, high-quality OOD training data.

### 6.3  Comparison between NLP and CV in OOD Detection

OOD detection is an active research field in CV communities (Yang et al., 2021) and comprehensive OOD detection benchmarks in CV are constructed (Yang et al., 2022). A few OOD detection approaches for NLP tasks are remolded from CV research and thus these approaches share a similar design. However, NLU tasks

have different characteristics compared to CV tasks. For example, models in NLP need to tackle discrete input spaces and handle complex output structures. Therefore, additional efforts should be paid to develop algorithms for OOD detection in NLP. Although this paper mainly focuses on NLP tasks, it is beneficial to give more discussion about the OOD detection algorithms designed for NLP and CV tasks. Specifically, we summarize the differences in OOD detection between NLP and CV in the following three aspects:

**Discrete Input**   NLP handles token sequences that lie in discrete spaces. Therefore distorting ID samples in their surface space (Ouyang et al., 2021; Choi et al., 2021; Shu et al., 2021) produces high-quality OOD samples if a careful filtering process is designed. On the contrary, CV tackles inputs from continuous spaces, where it is hard to navigate on the manifold of the data distribution. Du et al. (2022b;a) showed OOD synthesizing in the pixel space with a noise-additive manner led to limited performance.

**Complex Output**   Most OOD detection methods in CV are proposed for $K$-way classification tasks. However, in NLP, conditional language generation tasks need to predict token sequences that lie in sequentially structured distributions, such as semantic parsing (Lukovnikov et al., 2021), abstractive summarization, and machine translation (Ren et al., 2023). Hence, the perils of OOD are arguably more severe as (a) errors may propagate and magnify in sequentially structured output, and (b) the space of low-quality outputs is greatly increased as arbitrary text sequences can be generated. OOD detection methods for these conditional language generation tasks should consider the internal dependency of input-output samples.

**Contextual Information**   Some datasets in NLP contain contextual information. It is important to properly model this extra information for OOD detection in these tasks. For example, STAR (Mosig et al., 2020) is a multi-turn dialogue dataset, and effective OOD detectors should consider multi-turn contextual knowledge in their modeling process (Chen & Yu, 2021). However, most CV models only consider single images as their inputs.

## 6.4   Future Research Challenges

**OOD Detection and Domain Generalization**   In most practical applications, we are not only interested in detecting OOD inputs that are semantically shifted, but also required to build more robust ID classifiers that can tackle covariate shifted data (Yang et al., 2021). We believe there are opportunities to tackle problems of OOD detection and domain generalization jointly. Recent work in CV also shows that OOD detection and OOD generalization can be optimized in a unified framework (Bai et al., 2023). Future research opportunities can be explored to equip OOD detectors with better text representation extractors. Both new task design and algorithm development can be investigated.

**OOD Detection with Extra Information Sources**   Humans usually consider OOD inputs easily distinguishable because they can access external information besides plain texts (e.g., images, audio, and videos). OOD detectors are expected to perform better if we can equip them with inputs from different sources or modalities. Although various works are proposed to model each single information source, such as text or image, recent works have only begun to explore combining different sources and modalities (Ming et al., 2022; Ming & Li, 2023). These works have shown significant performance improvements by leveraging vision-language models for OOD detection. We also envision future research to equip OOD detectors with external knowledge, such as structured knowledge graphs. Also, note that this research direction still lies in the scope of our taxonomy shown in Figure 1 since these extra information sources can be either OOD or ID.

Moreover, Internet search engines are common approaches for humans to obtain external knowledge (Komeili et al., 2022). More research opportunities can be explored to build Internet-augmented OOD detectors that can utilize rich and updated knowledge yielded by search engines to enhance the OOD detection performance.

**OOD Detection and Lifelong Learning**   All previous approaches focus on detecting OOD inputs so that we can safely ignore them. However, OOD inputs usually represent new tasks that the current system does not support. Systems deployed in an ever-evolving environment are usually expected to continuously learn from these OOD inputs (without a full re-training) rather than ignoring them (Liu & Mazumder, 2021). However, humans exhibit outstanding abilities in learning new tasks from OOD inputs. We believe OOD

detectors are essential components in a lifelong learning system, and it is helpful to combine OOD detection with a downstream lifelong learning process to build stronger systems. Specifically, a possible scenario is to present a subset of detected OOD samples to human annotators and apply a lifelong learning algorithm to absorb these annotations without re-training the original model. Future works can be carried out to integrate these processes to pursue more human-like AI systems (Kim et al., 2022; He & Zhu, 2022).

**Theoretical Analysis of OOD Detection**   Despite impressive empirical results that OOD studies have achieved, theoretical investigation of OOD detection is far behind the empirical success (Morteza & Li, 2022; Fang et al., 2022). We hope more attention can be paid to theoretical analysis for OOD detection and provide insights to guide the development of better algorithms and applications.

## 7   Conclusion

In this survey, we provide a comprehensive review of OOD detection methods in NLP. We formalize the OOD detection tasks and identify the major challenges of OOD detection in NLP. A taxonomy of existing OOD detection methods is also provided. We hope this survey helps researchers locate their target problems and find the most suitable datasets, metrics, and baselines. Moreover, we also provide some promising directions that can inspire future research and exploration.

## Limitations

There are several limitations of this work. First, this survey mainly focuses on OOD detection approaches for NLP domains. Despite the restrictive scope, our work well complements the existing survey on OOD detection in CV tasks, and hence will benefit a well-targeted research community in NLP. Second, some OOD detection methods mentioned in this paper are described at a high level due to space limitations. We include details that are necessary to outline the development of OOD detection methods so that readers can get a comprehensive overview of this field. Our survey provides an elaborate starting point for readers who want to dive deep into OOD detection for NLP. Moreover, The term "OOD detection" has various aliases, such as "Anomaly Detection", "Outlier Detection", "One-class Classification", "Novelty Detection", and "Open Set Recognition". These notations represent similar tasks with subtle differences in detailed experiment settings. We do not extensively discuss these differences due to space limitations. Readers can refer to other papers for more detailed discussions (Yang et al., 2021). Finally, we do not present any new empirical results. It would be helpful to perform comparative experiments over different OOD detection methods (Yang et al., 2022). We leave this as future work.

## Ethics Statement

This work does not present any direct ethical issues. In this survey, we provide a comprehensive review of OOD detection methods in NLP, and we believe this study leads to direct benefits and societal impacts, particularly for safety-critical applications.

## Acknowledgement

Yixuan Li is supported by the AFOSR Young Investigator Program under award number FA9550-23-1-0184, National Science Foundation (NSF) Award No. IIS-2237037 & IIS-2331669, Office of Naval Research under grant number N00014-23-1-2643, and faculty research awards/gifts from Google and Meta.

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

## A    Surveying Process

In this appendix, we provide more details of how we select papers for our survey. Specifically, the selected paper follows at least one criterion listed below:

1. Peer-reviewed papers published in Top-tier NLP venues, such as ACL, EMNLP, NAACL, AAAI, and IJCAI.

2. Peer-reviewed papers that have a significant impact on the OOD detection area. These papers are not necessarily limited to NLP tasks.

3. Papers that are highly cited in the OOD detection area.

4. Most recently published papers that make a non-trivial contribution to OOD detection, such as methods, datasets, metrics, and theoretical analysis.

5. Papers that initiate each research direction in the OOD detection area.

| Dataset | Classes | #ID | #OOD | Papers that use this dataset (Selected) |
|---|---|---|---|---|
| CLINC150 (Larson et al., 2019) | 150 | 22,500 | 1,200 | (Zhang et al., 2021; Zhan et al., 2021; Lang et al., 2022; Zhou et al., 2022b) |
| Banking (Casanueva et al., 2020) | 77 | 13,083 | 0 | (Zhang et al., 2021; Zhan et al., 2021; Lang et al., 2022; Zhou et al., 2022b) |
| StackOverflow (Xu et al., 2015) | 22 | 20,000 | 0 | (Zhang et al., 2021; Zhan et al., 2021; Lang et al., 2022; Zhou et al., 2022b) |
| STAR (Mosig et al., 2020) | 150 | 27,510 | 1,594 | (Chen & Yu, 2021; Lang et al., 2023) |
| ROSTD (Gangal et al., 2020) | 12 | 43,323 | 4,590 | (Chen & Yu, 2021; Podolskiy et al., 2021) |

Table 1: More detailed information of various common datasets for OOD detection. # indicates the total number of samples.

## B  More details of Datasets

Table 1 provides more detailed information on various common datasets for OOD detection, regarding the total number of classes, the data size of ID and OOD samples, and selected papers using these datasets.

## C  More details of Metrics

Table 2 provides more detailed information on various metrics for OOD detection, regarding whether to consider ID performance, frequency of use, and applications.

| Metric | Definition | Whether to consider ID performance | Frequency of use | Applications | Papers that use this metric (Selected) |
|---|---|---|---|---|---|
| AUROC | Area under the Receiver Operating Characteristic curve | No | Very Frequent | NLP, CV, ML | (Hendrycks & Gimpel, 2017; Hendrycks et al., 2019a;b; Lee et al., 2018a) |
| AUPR-IN | Area under the Precision-Recall curve (ID samples as positive) | No | Frequent | NLP, CV, ML | (Lee et al., 2018a; Zheng et al., 2020; Shen et al., 2021) |
| AUPR-OUT | Area under the Precision-Recall curve (OOD samples as positive) | No | Frequent | NLP, CV, ML | (Lee et al., 2018a; Zheng et al., 2020; Shen et al., 2021) |
| FPR@$N$ | Value of FPR when TPR is $N\%$ | No | Not Frequent | NLP, CV, ML | (Lee et al., 2018a; Zheng et al., 2020; Shen et al., 2021) |
| F1 | Macro F1 score over all testing samples (ID+OOD) | Yes | Very Frequent | NLP | (Xu et al., 2019; Zhan et al., 2021; Shu et al., 2021; Zhou et al., 2022b) |
| Acc | Accuracy score over all testing samples (ID+OOD) | Yes | Very Frequent | NLP | (Zhan et al., 2021; Shu et al., 2017; 2021; Zhou et al., 2022b) |

Table 2: More detailed information of various metrics for OOD detection.

