# OpenReview forum: "A Survey on Out-of-Distribution Detection in NLP"
_TMLR — Accepted by TMLR_

### Review · Reviewer_3Qm5 · 2023-10-27

**Summary Of Contributions:**

This work is a survey on OOD detection for NLP. They start with a definition of OOD and relation to other areas. Then, a categorization of methods for OOD is given based on the amount of OOD data and ID labels, with a short description of the methods given. The standard datasets/tasks and metrics are also given. It concludes with a discussion and future challenges.

**Audience:**

Yes

**Claims And Evidence:**

Yes

**Requested Changes:**

Major
- Add section addressing LLMs

Minor
 - In intro, to better motivate problem, give more examples of why detecting OOD samples is important besides TOD
- In Fig 1, why are there are no methods for Extensive OOD data, Few OOD data, and Others? And Fig 1 can look better with less white space - maybe flip the diagram 90 degrees?
- In section 4, why does “Conditional Language Generation Tasks” require OOD detection?

**Strengths And Weaknesses:**

Strength - As someone not familiar with the field of OOD detection, I found this survey easy to follow while giving a good picture of OOD. The categorization based on the amount of OOD data and the amount of ID labels seem reasonable. Also, the descriptions of various methods are concise yet easy to understand. I also appreciate the last 2 sections on standard datasets/tasks and metrics. I found the survey was clearly written and gave a good roadmap of OOD detection for NLP - which is the purpose of a survey.

Weakness - The question I had in the back of my mind as I was reading this paper was “What are the implications of OOD in the ChatGPT era? The survey focuses on models and tasks that are more constructed for the pre ChatGPT era, but I think as a survey submitted now, it should also address LLMs. For example, what is out-of-domain for ChatGPT? If all NLP tasks are treated as text to text, than what is an unknown test label according to definition 2? How do LLMs help or not help the OOD detection problem?

---

> ### Author Response · Authors · 2023-11-09
> **Author response to Reviewer 3Qm5**
>
> We sincerely thank the reviewer for the time and effort in reviewing our paper and providing valuable feedback.
>
> We strongly agree that the survey will benefit from discussing the impacts of Large Language models (LLMs) on OOD detecttion. We think OOD detection should consider the influences of LLMs in two different aspects: **1.** LLMs encode a wealth of world knowledge and these knowledge can be used to help existing OOD detection approachs, such as generating text descriptions of in-domain class names [1] [2] and data augmentation [3]; **2.** In the LLMs era, NLP tasks can be treated as "text to text", which classify the next token in an output sequence by lanuage models [4]. LLMs may suffer even worse degradation on OOD inputs (e.g., different distribution from the training data) as the prediction is done auto-regressively over many steps. This is also described in Section 4 of this survey. Thus, novel OOD detection approaches are encouraged to be developed for this new issue. We will make sure to address this suggestion in our next revision.
>
> [1] Sachit Menon and Carl Vondrick. Visual classification via description from large language models. ICLR, 2023.
>
> [2] Dai, Yi and Lang, Hao and Zeng, Kaisheng and Huang, Fei and Li, Yongbin. Exploring Large Language Models for Multi-Modal Out-of-Distribution Detection. EMNLP, 2023.
>
> [3] Dai, Haixing and Liu, Zhengliang and Liao, and others. Chataug: Leveraging chatgpt for text data augmentation. Preprint, 2023.
>
> [4] Ren, Jie and Luo, Jiaming and Zhao, Yao and Krishna, Kundan and others. Out-of-distribution detection and selective generation for conditional language models. ICLR, 2023.

---

### Review · Reviewer_6Y3y · 2023-11-02

**Summary Of Contributions:**

This paper presents a survey of techniques for out-of-distribution (OOD) detection in NLP. It focuses specifically on settings where a semantic shift in distribution is detected; that is, the label space (as opposed to the input space) has changed. It provides a taxonomy of settings, depending on whether OOD data or labeled in-distribution (ID) data is available, and then goes on to briefly describe methods from each leaf in the taxonomy. It closes with a discussion of differences between OOD detection for vision and NLP and sketches some future research challenges.

**Audience:**

No

**Claims And Evidence:**

No

**Requested Changes:**

I think it’s very hard to come up with a list of requested changes for a survey paper. Were I writing something like this, I think I’d try to describe one key method from each of the major categories in depth, and then describe the others in contrast to that key method.

I would also suggest that the authors lead with some concrete examples of cases where OOD detection is necessary for NLP, and not leave that to be inferred from the Data section.

I will request that the authors clean up their citations. It’s very important that any citation not used as a noun in the text be completely enclosed in brackets. Also, some of the bibliography entries (the BERT one, for instance) are not formatted correctly.

**Strengths And Weaknesses:**

Strengths:

As a narrative inventory of papers on OOD detection in NLP, this paper seems very thorough and up-to-date.

Weaknesses:

I don’t see survey papers listed among the contribution types in https://jmlr.org/tmlr/editorial-policies.html. Even if we assume that pure surveys are welcome, I don’t think this survey has sufficient value-add to be of interest to the TMLR readership. As mentioned above,  the main contributions here are the inventory of papers and the taxonomy used to organize them. What’s missing is any sort of meaningful explanations of the methods themselves or an attempt to draw connections or generalizations among them. For a survey to truly add value, it should not only provide a jumping-off point through an annotated bibliography (which this survey does nicely) but it should also help the reader gain some insight into the commonalities between various methods.

I did not feel like I understood any one method particularly well after reading this. The paper reads like a related work section of a larger paper - with each method getting exactly one sentence. I also found the actual NLP problems being solved to be very abstract and hard to grasp – the closest I came to clarity on this was when the various dialogue-related datasets were being described.

In terms of clarity concerns: The paper on several occasions calls out the importance of complex output structures (trees or sequences) as a defining characteristic of NLP applications of OOD detection. However, this seems to run contrary to the paper’s focus on problems with semantic drift – where the label space changes as opposed to the input space. Surely this semantic drift is difficult to define when dealing with generation tasks like summarization or machine translation? I was hoping to hear more discussion of how this concept of drift even applies in such wide-open output settings, once we move beyond multi-class classification.

---

> ### Comment · Action_Editor_QBan · 2023-11-08
> **Clarification about survey papers**
>
> Hi,
>
> Thank you very much for your review. While indeed the editorial policies page is a bit vague on surveys, they are very welcome, as clarified in the TMLR FAQ: https://jmlr.org/tmlr/faq.html
>
> > Q: Does TMLR accept survey papers?
>
> > A: Yes. Authors should make sure to emphasize the contributions made by the survey. Ideally, we want survey papers that draw new, previously unreported connections between several pieces of work in an area, and/or that clearly highlight trends in the area and/or suggest currently open problems. It should also be noted that if a submission has more than 12 pages of main content, then TMLR's normal short review timeline will not be enforced.
>
> I kindly ask you (the reviewer) to keep this in mind during the rest of the process and to please update your review accordingly.
>
> For the authors: you are not expected to dedicate space to respond to the welcomeness of surveys in general. (feel free however to justify the value and contributions of your survey.)
>
> Thanks!

---

> > ### Comment · Reviewer_6Y3y · 2023-11-08
> > **Acknowledged**
> >
> > Just letting you and the authors know that I've seen this, and I'll be sure to assess the contributions of the survey rather than the welcomeness of surveys in general.

---

> ### Author Response · Authors · 2023-11-09
> **Author response to Reviewer 6Y3y**
>
> We sincerely thank the reviewer for the time and effort in reviewing our paper and providing valuable feedback.
>
> We address each of the pointed weaknesses below:
> 1. "I don’t see survey papers listed among the contribution types":
> Kindly note that survey papers have been accepted by TMLR(https://jmlr.org/tmlr/papers/).
> 2. "I don’t think this survey has sufficient value-add to be of interest to the TMLR readership":
> For readers not familiar with the field of OOD detection in NLP, this survey gives a good picture and roadmap of OOD detection. Meanwhile, relevant algorithms are categorized into three distinct classes based on the data they utilized, which help readers use suitable OOD detection methods in practice. Moreover, comparisons betwen NLP and CV in OOD detection are discussed in this survey, which raise some interesting points that can be explored in future research.
> 3. "I did not feel like I understood any one method...with each method getting exactly one sentence":
> Kindly note that we have introduced the shared characteristics of sub-methods in each direction before diving into model details (For example, see the beginning paragraph of Section 3.2.1, Section 3.2.2, Section 3.2.3).
> 4. "more discussion of how this concept of drift even applies in such wide-open output settings, once we move beyond multi-class classification":
> Thanks for the suggestion. In the large language models (LLMs) era, NLP tasks can be treated as "text to text" beyond multi-class classification. In this setting, LLMs are trained to classify the next token in an output sequence [1]. LLMs may suffer even worse degradation on OOD inputs (e.g., different distribution from the training data) as the prediction is done auto-regressively over many steps. Thus, novel OOD detection approaches are encouraged to be developed for this new issue. We promise to include a new discussion on "Addressing LLMs".
>
> [1] Ren, Jie and Luo, Jiaming and Zhao, Yao and Krishna, Kundan and others. Out-of-distribution detection and selective generation for conditional language models. ICLR, 2023.

---

### Review · Reviewer_4Z5E · 2023-11-06

**Summary Of Contributions:**

This paper delves into the domain of out-of-distribution (OOD) detection, an imperative aspect for ensuring the reliability and safety of machine learning systems when deployed in real-world scenarios. Over recent years, substantial advancements have been achieved in this field, and this paper offers an overview of these developments with emphasis on natural language processing (NLP) approaches.

The paper provides a formal definition of OOD detection to build a theoretical foundation, and then delves into the exploration of related fields. It categorizes the relevant algorithms into three distinct classes based on the data they utilize: (1) those with OOD data readily available, (2) those with OOD data not available but in-distribution (ID) labels are accessible, and (3) those operating with both unavailable OOD data and ID labels. The paper also discusses some relevant OOD datasets, common OOD application tasks in literature, and metrics pertinent to OOD detection. It also lays out potential avenues for future research in this area.

**Audience:**

Yes

**Claims And Evidence:**

Yes

**Requested Changes:**

**Introduction**
* “Natural language processing systems deployed in the wild often encounter out-of-distribution (OOD) samples that are not seen in the training phase.” In what way? Can you please elaborate a little? Can you add a sentence or 2 of how LLMs are currently handling it (or not handling)?
* Is the italicization intended and/or needed?
* It is mentioned that this study is comprehensive. However, if the focus of the paper is only on semantic shift and not non-semantic (e.g. OOD domain generalization), would calling it comprehensive be too strong of a claim? If so, maybe rephrase?

**Section 2**
* For zero-shot learning, I'm not sure I understood the contrast you're trying to draw between OOD detection and Wang et al. Can you please elaborate? IIUC we could also use zero-shot to setup the model to label and flag the unseen class (e.g. zero-shot prompting).
* Consider adding Transfer Learning as a related area?

**Section 3**
* 3.1: Since the fundamental aspect of the paper is to present a taxonomy based on data OOD data availability, I think it needs a table to show relevant datasets in the categories mentioned, the paper or techniques using those datasets, size of the datasets (especially their OOD splits or %). Without this, I feel there's a bit of a disconnect as the section focuses on a dataset based taxonomy, but lists models and prior work based on the techniques.
* Is it possible to have a brief discussion of why methods from 3.1.1 don't work in 3.1.2 or vice versa? Or are there techniques that work for both scenarios?
* 3.2.1: IIUC Hendrycks et al 2020 uses the model confidence scores to set up a threshold and then gets the FAR95 on the OOD detection dataset. For taxonomic characterization purposes, to me it seems similar to the binary classification/thresholding techniques from 3.1.1. Would that be a correct way to interpret this? If yes, consider moving Hendrycks 2020 or refining the definitions of those sections a bit more to demarcate better?
* Similar to the above point, Hendrycks 2020 also seems to fall in 3.2.1b output based detecting (as would other techniques in 3.2.1a) since it's essentially using a thresholding function. Is that a correct way to interpret? If yes, consider refining the definitions to disambiguate?

**Section 6**
* While the paper mentions an interesting discussion topics (OOD detection in NLP vs CV), it is unable to elaborate on it further (probably due to limited space or the broad focus area of the paper). I was wondering: is there additional material in this section the authors were considering adding (e.g. datasets, benchmarks, etc)? If yes, is it possible to address this in the appendix while still following TMLR length guidelines?
* The focus of section 6.3 seems a bit too broad and a bit disconnected with the paper (i.e. OOD detection). Is the discussion of DG be relevant here? Also, "OOD Detection with Extra Information Sources" seems to be about adding more context to an example in a dataset. That context can be ID or OOD. In that scenario, wouldn't this new contextual dataset again reduce to the ones mentioned in section 3? Also regarding "OOD Detection and Lifelong Learning", I think practically deployed classifiers usually have this re-training loop: a subset of OOD detected data is presented to human annotators who can reclassify them appropriately and retrain the model with them. Could you elaborate a bit more on what you mean by the future research aspect of this, or add any relevant reference (e.g. RL system to continually improve OOD detector)?

**Strengths And Weaknesses:**

**Strengths**
* Since this is a literature review of SoTA OOD detection theory and practical techniques, it references relevant and significantly impactful research.
* Clearly states some of the limitations of the work
* Section 6.2 discussion of the comparison betwen NLP and CV in OOD detection raises some interesting points that can be explored in future research. Nature of NLP tasks are still limited by somewhat straightforward modality of language (i.e. same type of text tokens roll up to higher level concepts and so on). For images, the variation in the nature of these tokens is immense (pixel art, vs ASCII art vs Abstractionist vs DSLR photography).

**Weaknesses**
* The paper astutely identifies availability of data as a good feature to use to characterize OOD detection solutions. However, I think it needs to focus a bit more on that aspect. Section 4 - which I think should have been better presented via tables, and more clearer description and discussion of the datasets, the benchmarks they serve, and the top solutions for them - is currently a bit tricky to follow. Similarly for Section 5, I would have liked some more discussion on why these metrics were chosen, and pros and cons of one vs the other (preferably in tabular format). Similarly for section 3. While it lays out the methodology for defining the taxonomy, but the listed research mostly focuses on the techniques, and not so much on how those techniques are influenced by the data category they're in. As a result, these sections read like a writeup on a collection of relevant papers and not an authoritative survey as the authors intend.

---

> ### Author Response · Authors · 2023-11-09
> **Author response to Reviewer 4Z5E**
>
> We sincerely thank the reviewer for the time and effort in reviewing our paper and providing valuable feedback.
>
> We address each of the pointed weaknesses below:
> 1. "more clearer description and discussion of the datasets...how those techniques are influenced by the data category they're in"：
> You are right. We will add a new table to better present the characteristics of datasets and their suitable methods.
>
> 2. "I would have liked some more discussion on why these metrics were chosen, and pros and cons of one vs the other"：
> Thanks for the suggestion. We promise to provide more discussions on metrics in a new table.
>
> 3. "for section 3...these sections read like a writeup on a collection of relevant papers"：
> Kindly note that we have introduced the shared characteristics of sub-methods in each direction before diving into model details (For example, see the beginning paragraph of Section 3.2.1, Section 3.2.2, Section 3.2.3). We will show the intrinsic relationship between techniques and the data available in our revised paper.

---

### Author Response · Authors · 2023-11-19
**Paper revision**

We firstly extend our heartfelt appreciation to the esteemed reviewers for generously dedicating their time and providing invaluable feedback. We have diligently taken into account all the suggestions put forth in the reviews and subsequently revised our manuscript. To facilitate convenience and transparency, we present a comprehensive changelist and a corresponding legend, clearly delineating the modifications made to address each reviewer's suggestions. We eagerly welcome any additional thoughts or suggestions to further refine and augment our survey.

*Legend for reviewer reference:*

**Reviewer 4Z5E**: Changes #1, 2, 4, 7, 8

**Reviewer 6Y3y**: Changes #1, 5, 6, 9

**Reviewer 3Qm5**: Changes #1, 3, 5, 6

*Changelist*:

1.**(Page 1, Section 1)** Add more examples to show the importance of detecting OOD samples, even with LLMs.

2.**(Page 2, Section 2)** Elaborate on zero-shot learning, and add transfer Learning as a related area.

3.**(Page 3, Section 3)** Flip Figure 1's diagram 90 degrees to look better.

4.**(Page 4, Section 3.1.2, Section 3.2.1.b)** Add a brief discussion of the relationship between methods with few OOD data and methods with extensive OOD data; Hendrycks 2020 falls in 3.2.1b output based detecting.

5.**(Page 7, Section 4)** Remove the paragraph about "Conditional Language Generation Tasks" because its content is now described in the newly added secton "Large Language models for OOD Detection".

6.**(Page 9, Section 6.2)** Add Section "Large Language models for OOD Detection" addressing LLMs.

7.**(Page 9, Section 6.3, Section 6.4)** Add recently built comprehensive OOD detection benchmarks in CV; Improve the discussion for future work.

8.**(Page 19, Appendix B)** Add a table of more information on common OOD detection datasets.

9.**(All Pages)** Clean up citations.

---

### Decision · Action_Editor_QBan · 2023-12-12

**Recommendation:** Accept with minor revision

**Comment:**

This survey contains a lot of good work that could benefit readers working in the area in finding relevant literature. While the reviewers find the survey to be not particularly insightful structurally, it is deemed valuable as an inventory of similar approaches and it would be a benefit to the TMLR readers. I recommend acceptance after a minor revision: thoroughly fixing the citation formatting as previously requested by reviewers (some inconsistencies remain at least in section 6.4)

**Audience:**

Yes. Reviewers agree that this survey can help navigate the surveyed field.

**Claims And Evidence:**

Yes. The paper surveys the field of out-of-distribution detection in NLP; reviewers find it a thorough inventory and a fairly useful attempt to structure the field.